# Optimization of Production Parameters for Probiotic Lactobacillus Strains as Feed Additive

**DOI:** 10.3390/molecules24183286

**Published:** 2019-09-09

**Authors:** Hao Ren, Jürgen Zentek, Wilfried Vahjen

**Affiliations:** Institute of Animal Nutrition, Freie Universität Berlin, Königin-Luise-Str. 49, 14195 Berlin, Germany; Juergen.Zentek@fu-berlin.de (J.Z.); wilfried.vahjen@fu-berlin.de (W.V.)

**Keywords:** probiotic, optimization procedure, freeze-drying, response surface method, in-feed stability

## Abstract

In animal nutrition, probiotics are considered as desirable alternatives to antibiotic growth promoters. The beneficial effects of probiotics primarily depend on their viability in feed, which demands technical optimization of biomass production, since processing and storage capacities are often strain-specific. In this study, we optimized the production parameters for two broiler-derived probiotic lactobacilli (*L. salivarius* and *L. agilis*). Carbohydrate utilization of both strains was determined and preferred substrates that boosted biomass production in lab-scale fermentations were selected. The strains showed good aerobic tolerance, which resulted in easier scale-up production. For the freeze-drying process, the response surface methodology was applied to optimize the composition of cryoprotective media. A quadratic polynomial model was built to study three protective factors (skim milk, sucrose, and trehalose) and to predict the optimal working conditions for maximum viability. The optimal combination of protectants was 0.14g/mL skim milk/ 0.08 g/mL sucrose/ 0.09 g/mL trehalose (*L. salivarius*) and 0.15g/mL skim milk/ 0.08 g/mL sucrose/ 0.07 g/mL (*L. agilis*), respectively. Furthermore, the in-feed stabilities of the probiotic strains were evaluated under different conditions. Our results indicate that the chosen protectants exerted an extensive protection on strains during the storage. Although only storage of the strains at 4 °C retained the maximum stability of both *Lactobacillus* strains, the employed protectant matrix showed promising results at room temperature.

## 1. Introduction

The development of alternatives for antibiotic growth promoters in livestock has been of global interest in the field of animal nutrition after their ban in many countries [1,2,3]. Probiotics have shown beneficial effects in the field of animal nutrition such as improved weight gain, development of a beneficial intestinal microbiota, and enhancement of the immune system in farm animals [4,5,6]. Most probiotic are bacteria, but there are also few non-bacteria microorganisms like yeast that belong to the probiotic family [7]. As an important member of lactic acid bacteria, *Lactobacillus* has become one of the most commonly used probiotic species among all probiotic species [8]. Health promotion by lactobacilli, which are generally regarded as safe (GRAS), makes them particularly interesting as a zootechnical additive [9,10].

A sufficient number of viable probiotic cells is a prerequisite for their successful impact in the animal [11]. In order to commercialize probiotics, timesaving and cost-effective methods to increase bacterial cell yield during the production progress are necessary [12]. Among other parameters, biomass production can be improved by adjusting growth factors (e.g., substrates, pH, incubation time) to optimize biomass production [13,14,15]. Another fundamental factor is the cost intensive fermentation especially of anaerobes, which negatively affects the scale-up of biomass production [16]. This topic has been investigated in several studies, but with a limited number of candidate species [17].

The preparation of probiotic products calls for reasonable cell stability during the manufacturing process. Among various techniques, drying methods are commonly used for the preservation and ease of handling of microorganisms [18]. Freeze-drying has been widely applied to bacteria that exhibit high stability against low temperatures [19]. However, stress factors such as very low freezing temperatures or dehydration during freeze-drying can cause undesirable loss of viability for some probiotic strains [20,21]. Due to this, a variety of cryo-protectants have been developed to increase the viability of probiotic bacteria during the freeze-drying procedure [22,23]. Protectants such as skim milk, whey proteins, sugars, or other bio-polymers were studied mostly as combinations for synergistic protective effects with other protectants [24,25]. The classical one-variable-at-a-time approach (OVAT) strategy was deemed more time-consuming. It ignores the interaction between functioning factors, which might lead to the confusion and bias of results [26]. Thus, the response surface method has become one of the most used optimization approaches to create the best conditions with a minimum number of experiments [25]. Among different optimization procedures, the Box-Behnken Design (BBD) has been shown to be superior to 3-level full factorial designs and is, thus, being used in response surface modelling [27,28]. Furthermore, results indicated that cryo-protectants might work in a strain-specific manner and, thus, optimization may rely on particular protective systems for a given strain.

Viability and activity of probiotics during storage are critical criteria for both the manufacturer and customer [29]. Storage conditions affect survival of bacterial cells [30] and can even influence the functionality of the probiotic such as stress resistance or capacity of epithelial adhesion without changing cell viability [31]. Most studies report on the storage stability of probiotics as a sole objective. However, in-feed stability is of prime importance, but is seldom reported.

In a previous study, two *Lactobacillus* strains (*L. salivarius*, *L. agilis*) were isolated from broiler intestinal samples (unpublished data). These strains were tested for their applicability as a probiotic additive for poultry. The current study determined the most economical and feasible procedure to produce those probiotic strains as feed additive. Furthermore, different factors regarding biomass production, survival during lyophilization, and in-feed stability of storage were evaluated.

## 2. Results

### 2.1. Metabolic Fingerprints of the Lactobacillus Strains

The results for the BIOLOG^®^ AN plates are shown as a heat map in Figure 1. The *L. salivarius* strain showed a broader carbohydrate utilization spectrum than the *L. agilis* strain. The highest metabolic activity for the *L. salivarius* strain was observed for maltose, raffinose, sucrose, and glucose, while the *L. agilis* strain metabolized mannose, glucose, L-lactic acid, and mannitol as preferred carbohydrate substrates followed by mannitol, lactic acid, mannose, glucose, maltose, sucrose, maltotriose, lactose, melibiose, raffinose, sorbitol, and lactulose. Taking cost and easy-availability of those substrates into consideration, mannitol, mannose, maltose, sucrose, melibiose, and sorbitol were selected for further evaluation of boosting effects on lactobacilli growth.

### 2.2. Booster Effects of Selective Carbon Sources on Biomass Production

The addition of sucrose and sorbitol to the basal medium led to a significantly increased number of viable cells for strain *L. salivarius* after 12 h of incubation, while the addition of mannose revealed a booster effect on bacterial growth for the strain *L. agilis* (Table 1). Extension of cultivation time to 24 h showed that, all incubations exhibited lower viable cell numbers than after 12 h, except for incubations in the basal medium. The lowest viable cell numbers were observed after 48 h of incubation (see Table 1), whereas the biomass in all experimental groups decreased to a level significantly lower than in the MRS medium.

When comparing all cultivation situations, the incubation of 12 h with the addition of sucrose significantly increased the biomass yield of strain *L. salivarius* (*p* = 0.05). Although the addition of mannose did not significantly increase the biomass yield of *L. agilis* (*p* = 0.127), it ascertained that shortening the cultivation time still yielded high biomass for both strains. These two substrates were used in further tests to increase the biomass yield for *L. salivarius* and *L. agilis*, respectively.

### 2.3. Effect of Aerobic or Anaerobic Incubation on Biomass Production

The tolerance of both strains to oxygen was evaluated by growth under aerobic or anaerobic conditions. Compared to aerobic conditions, the *L. salivarius* strain demonstrated numerically increased biomass under anaerobic conditions (11.97 ± 11.40 log CFU/L anaerobic vs. 11.90 ± 10.74 log CFU/L aerobic). There was also no significant difference in the biomass of strain *L. agilis* between anaerobic incubation and aerobic incubation (12.01 ± 11.17 log CFU/L anaerobic vs. 12.02 ± 11.07 log CFU/L aerobic).

### 2.4. Lyophilization and Optimization of Lyo-Protectants

With the purpose of defining the best survival of the strains after lyophilization, a total of 17 experiments with appropriate combinations of the three chosen protectants (skim milk, sucrose, and trehalose) were performed, according to the Box-Behnken Design (BBD).

Both actual and predicted responses of the strains with a different combination of factors were used for the establishment of a quadratic model (Appendix A). The ANOVA (Analysis of variance) fitted quadratic polynomial model is presented in Table 2. Data in both models were different with a high significance. The value of the determination coefficient also confirmed the goodness of fit for the polynomial model. Coefficients are the effects of each factor. By interpreting the results, it is possible to define the factor or factor combinations that have higher influence. The significances of all coefficients are shown in Table 2. In the current case, most linear coefficients, square coefficients, and interaction coefficients of the *L. salivarius* model (X_1_, X_2_, X_1_X_2_, X_2_X_3_, X_1_2, X_2_^2^, and X_3_^2^) and the *L. agilis* model (X_1_, X_2_, X_3_, X_1_X_2_, X_1_X_3_, X_2_^2^, and X_3_^2^) were significant model terms, which confirmed the validation of the model.

The fitted response surface plots and their corresponding contour plots for the survival of the strains after lyophilization are shown in Figure 2 and Figure 3. The diagnostic of the modelling demonstrated that all residuals of both responses were normally distributed as linearity, which validated the statistical assumption of the model (Appendix A). The predicted vs. actual value of survival of both *L. salivarius* and *L. agilis* are presented in Figure 4.

The optimal concentration for each variable was deduced from the software as 0.14 g/L skim milk, 0.08 g/L and 0.09 g/L trehalose for *L. salivarius*, and 0.15 g/L skim milk, 0.08 g/L, and 0.07 g/L trehalose for *L. agilis*, respectively. With the optimized formulation of cryo-protectants, the maximum survival of both *L. salivarius* and *L. agilis* could be demonstrated (Table 3).

### 2.5. Stability during In-Feed Storage

The stability of both strains was determined according to their time-dependent in-feed survival after mechanical mixing in the feed mill. The cryo-protectants showed no significant effects against feed processing, since no difference with or without protectants was observed for both strains (Table 4). The *L. salivarius* strain suffered only from a small numeric decrease in the cell numbers. Similarly, the protectants demonstrated no significant protection effect for the *L. agilis* strain. The refrigerated storage revealed slightly higher viability than storage at room temperature. Short-term storage (day 0–4) showed remarkable in-feed survival rates for both strains without differences of storage with or without cryo-protectants.

As to the mid-term storage (day 5–15), the survival of the *L. salivarius* strain with protectants under a refrigerated condition was higher than without protectants at day 15. However, the difference between the strain with protectants at room temperature and the strain without protectants at a refrigerated condition was not significant. Long-term storage for 28 days showed that the *L. salivarius* strain with protectants at a refrigerated condition exhibited a notably higher survival rate than under any other condition. When incorporated with protectants, the viability of the *L. agilis* strain was significantly higher on day 15 and 28. The details were shown in Table 4.

## 3. Discussion

The advantageous role of probiotics in human and animal health has been well accepted. The promising potential is increasingly used in animal nutrition [10]. Among the challenges toward the commercialization of probiotic products, the main factor is the delivery of adequate amounts of viable bacteria at the time of administration [32]. Thus, the optimization of production parameters for specific probiotic strains is of high importance. The current study investigated optimal and cost-effective preparation procedures to ensure a high yield of biomass and maximum in-feed stability of two probiotic strains that were isolated in a previous study. The efficiency of probiotic products is highly dependent on cell viability, since the mode of action of probiotics is conferred by living cells [33]. Thus, a prerequisite for a successful probiotic product is its stability throughout the processing and storage until delivery. Our present study aimed to investigate the optimal and cost-effective preparation procedure for two selected probiotic *Lactobacillus* strains. Aspects of biomass production, protection during lyophilization, and in-feed storage stability were investigated.

The utilization of substrates by lactobacilli is characterized by species-specific or strain-specific differences during growth [34]. To define the specific carbon source preferences of the probiotic *Lactobacillus* strains, the BIOLOG^®^ technology was employed in this study. The microtiter plate-based BIOLOG^®^ methodology is primarily used as a tool for identifying bacteria [35] and has also been used as a tool to compare the metabolic activity of microbial communities from different habitats [36]. The BIOLOG^®^ system is based on the reduction of a redox dye, which indicates bacterial utilization of substrates [37]. Thus, color development during growth not only indicates substrate use, but is also directly proportional to metabolic activity. This potential was used to rapidly identify the preferential substrate utilization of the two probiotic *Lactobacillus* strains. Substrate utilization varied as expected, which shows specific substrate preferences for each strain. After ranking by OD (optical density), the six top substrates were selected for further evaluation.

MRS (de Man, Rogosa, and Sharpe) medium was used in this study, because it is the most commonly used complete medium to allow growth of lactic acid bacteria [38]. The selected carbon sources were added as additional substrates to determine whether they would enhance cell growth on top of the already present glucose. Our results indicate that the addition of sucrose for *L. salivarius* and mannose for *L. agilis* shortened the exponential growth phase and yielded more biomass than with MRS alone.

*Lactobacillus* spp. are facultative anaerobes, but several species do not tolerate oxygen well [39,40]. Since aerobic cultivation has less energy and is cost intensive, economic advantages can be gained, if technical biomass production can be run under aerobic conditions [41]. Therefore, it was essential to know whether the selected probiotic strains grew equally well under an aerobic condition. As the two strains showed good oxygen resistance, they should be able to be cultured aerobically under large-scale technical conditions. This will lead to a more economic biomass production for those strains.

Extended incubation time (48 h) led to cell loss, which was likely subjected to the self-inhibition caused by accumulation of lactate or other end metabolites [42]. Therefore, biomass production was set to 12 h in the MRS medium supplied with booster substrates. Freeze-drying is one of the commonly employed techniques to produce viable bacterial cells for long-term storage [43,44]. However, a fraction of cells is lost during the lyophilization process because of ice crystal formation with subsequent damage to the viable cell [45]. To maintain viability, a variety of cryo-protectants have been developed to provide structural dry residues as support as well as to act as rehydration receptors [46]. Therefore, cryo-protectants also play an important role in the conservation of probiotic products, which lead to higher survival of probiotic strains [47,48].

Several studies addressed the generation of a protective medium for *L. salivarius* strains, but different methods and optimal media compositions were found in different studies [49,50,51]. This suggests that protective effects are strain-dependent. To our best knowledge, it is the first study on optimization of cryo-protection for *L. agilis*. Although protection might be strain-dependent, the beneficial action of skim milk for the *L. agilis* strains may also hold true for other *L. agilis* strains. Thus, future studies on *L. agilis* may also include skim milk as a cryo-protectant during optimization.

Multiple compounds in a cryo-protective mixture were often found to yield synergic effects [52]. Hence, three potential protective factors were used in this study, i.e., skim milk, sucrose, and trehalose. To better understand how the three factors interacted and to find the optimal working concentrations, the Box Behnken Design (BBD) for multivariate optimization schemes with simultaneously changed variables was applied to build a mathematical model with experimental data [27]. The most influencing factor for both strains was skim milk, which is consistent with other investigations [53]. Skim milk for protection of viable cells stabilizes bacterial cell membranes and enables an easier rehydration by creating a high surface porous structure [54]. Both sucrose and trehalose enhanced survival of the cells in addition to the protective effect of skim milk. A similar synergistic effect was reported previously for *Candida sake* cells. In that study, the single use of sucrose did not significantly increase cell viability, but protected the cells better, when skim milk was used during freeze-drying [55].

The protection of bacterial cells by disaccharides is generally attributed to their capacity to hydrate biological structures, which is referred to as a ‘water replacement hypothesis’ [56]. In studies on the activity of protective sugars, trehalose was shown to be the most effective compound for a range of lactic acid bacterial strains (*L. bulgaricus*, *L. acidophilus*, and *S. salivarius* etc.) [57]. In our case, trehalose did not act as a predominant factor, as demonstrated by a similar effect like sucrose. Between the tested lactobacilli, the *L. salivarius* strain was more dependent on trehalose. Not only the positive influence on viable biomass during the lyophilization, but also improvement of viability during storage has been reported for a range of protectants [58]. Several studies used skim milk, sucrose and trehalose alone or in combination [49,59,60]. To our knowledge, storage in a feed matrix is rarely tested for probiotics in animal nutrition. In one study, a mixture of *Bacillus* spp. was tested as liquid culture in prawn feed. Similar to our study, their results also indicated that probiotic *Bacillus* spp. strains were more stable at a lower temperature (4 °C). Nevertheless, the survival of their isolates at room temperature after 28 days was actually lower than in our study, which can be assigned to a lower stability of liquid cultures compared to dried powders [61].

Storage at 4 °C is not possible for animal feeds, as energy demands for tons of feed would be prohibitively high. Although the temperature exerted a significant impact on survival, it was evident that the combination of protectants enhanced the in-feed stability throughout storage. Furthermore, the *L. salivarius* strain also showed improved stability against physical mixing, when combined with cryo-protectants. On the contrary, the *L. agilis* strain seemed to be more tolerant against a physical force, since no significant changes were observed between cryo-protectants or non-protected feed samples. This corresponds to a report by Sadguruprasad and coworkers (2018) who found highly variable and strain-dependent storage effects on microorganisms [62]. However, the designated protectants in this study benefited the stability of both strains from short-term to mid-term storage when mixed and stored with feed.

## 4. Materials and Methods

### 4.1. Strains and Medium

The strains were isolated from broiler intestinal samples and taxonomically identified as *L. salivarius* and *L. agilis* by 16S rDNA sequencing. Both strains were stored in cryo stock at −80 °C. They were cultivated in de Man, Rogosa and Sharpe (MRS, Carl Roth GmbH + Co. KG, Germany) broth in anaerobic jars (Merck KGaA, Germany) with Anaerocult C (Merck KGaA, Germany) at 37 °C for 24 h. The inoculum was prepared fresh each time before use. MRS agar plates were used to determine the viable cell number after treatment.

### 4.2. Metabolic Fingerprint of Probiotic Lactobacillus Strains

BIOLOG^®^ AN plates (BIOLOG^®^ Inc., Hayward, CA, USA) were used to identify the substrate utilization pattern of the isolates [37]. The technology can also be used to determine substrate utilization patterns of microbial communities [63]. In the present study, the BIOLOG^®^ AN type plate was used to determine the carbohydrate preference of the *Lactobacillus* strains. The procedure followed the manufacturers’ guide with a minor modification. Both strains were inoculated in de Man, Rogosa and Sharpe medium (MRS, Carl Roth GmbH + Co. KG, Germany) and incubated in anaerobic jars (Merck KGaA, Germany) with Anaerocult C (Merck KGaA, Germany) overnight. The cultures were then washed with Phosphate Buffered Saline (PBS), pH 7.4, for three times and diluted to 10^7^ cells/mL. A total of 100µL bacterial suspension was pipetted into each well of BIOLOG^®^ AN plate in triplicate. The plates were incubated in anaerobic jars with Anaerocult C at 37 °C for 24 h and optical density was read with a microtiter plate reader (Tecan Infinite200Pro, Germany) at OD_590nm_.

### 4.3. Booster Effects of Additional Carbohydrate Sources on Biomass Production

Six carbohydrates (sucrose, maltose, mannitol, sorbitol, and melibiose) were selected as possibly beneficial for an increased biomass production of the two probiotic strains. The carbohydrates were added to MRS medium and supplemented with each of the selected additional substrates at a concentration of 1% (*w/v*) and each strain was inoculated into 100 mL of each carbohydrate-supplemented medium reaching a final inoculum of 10^6^ CFU/mL. After anaerobic cultivation at 37 °C for 12 h, 24 h, and 48 h, respectively, the resulting biomass was enumerated by plating.

### 4.4. Determination of Bacterial Growth under Aerobic or Anaerobic Conditions

Pre-cultures of both strains were prepared as described above. An inoculum of each strain was inoculated into 500 mL MRS medium with 10^5^ CFU/mL and incubated either in an anaerobic jar with Anaerocult C or in an aerobic incubator at 37 °C. After 12 h of incubation, the biomass of each culture was determined by plating.

### 4.5. Lyophilization and Optimization of Cryoprotectants

Pre-cultures were harvested after culturing under an aerobic condition at 37 °C for 12 h. Biomass was concentrated by centrifugation (10 min, 15,000 g, 4 °C) and resuspended in different protective media. Each medium contained combinations of sucrose, skim milk, and trehalose at different concentrations (see Appendix A). The suspensions were transferred into lyophilization boxes, incubated at −80 °C for 48 h, and dehydrated at −55 °C in a lyophilizer (LyoVac GT2, LC Didactic, Hürth, Germany) for 48 h. The freeze-dried biomass was ground into powder with a mortar and pestle and stored at 4 °C. The survival of the strain was determined by plating.

The optimization of cryoprotectants was performed using the response surface methodology [64], by which a response surface model was constructed for optimization with a sequential quadratic programming approach.

The survival of both lactobacilli was considered to be an individual response. The Box Behncken Design (BBD) with three factors (skim milk: X_1_, sucrose: X_2_, and trehalose: X_3_) and the software Design Expert 8.06 (Stat-ease Inc., Minneapolis, MN, USA) was used to analyze the survival data. The analytical procedure was referred to a study with minor modification [25]. A three-variable BBD with six replicates at the center point was selected to build the response surface models. The design is shown in the Appendix A. Analysis of variance (ANOVA) was performed to determine the post prediction and reproducibility of assessed combinations. The design was used to determine an optimal composition of protective medium by fitting the polynomial model on the basis of the response surface methodology [65].

### 4.6. In-Feed Stability of Probiotic Products

Both strains were prepared by lyophilization with or without cryo-protectants, as described above. A basal feed for broiler chicken was produced in mash form in the feed mill of the Institute of Animal Nutrition, Freie Universität Berlin (Appendix A). The probiotic products were homogenized in the feed with a feed mixer (5 kg) at an approximate concentration of 10^7^ CFU/g. The following treatments were applied to the mash feeds: with or without cryo-protectants at room temperature or 4 °C storage. All feed samples were stored for a maximum of 28 days. Subsamples (2 g) were drawn at 0, 1, 2, 3, 4, 15, and 28 days of storage and serially diluted in PBS (Phosphate buffered saline). Residual CFU/g of the strains was determined by plating. The in-feed survival rate was calculated as: survival rate [%] = CFU/g detected at day n post mixing (DPM_n_)/CFU/g before mixing (BM) ×100.

### 4.7. Statistical Analysis

All experiments were performed twice in triplicates. The results are presented as means ± standard deviation (SD). The Design Expert 8.06 software was used for the data analysis estimation of responses and prediction of optimized parameters by plotting response contours and surface graphs. Statistical significances of comparisons were assessed using one-way analysis of variance (ANOVA) or the Mann-Whitney test with the statistics software IBM SPSS (Version 22, SPSS Inc., Chicago, IL, USA).

## 5. Conclusions

In summary, two broiler-derived probiotic *Lactobacillus* strains (*L. salivarius* and *L. agilis*) were characterized for their preferred substrate utilization, biomass production, and oxygen tolerance as well as their optimal protective agents during freeze-drying and in-feed storage. The response surface methodology was employed to study the optimal composition of protective agents. The prepared probiotic products were supplemented into feed and, although viability decreased, more viable cells were recovered from samples with protectants. This study showed that optimal routines for lab-scale production, processing, and storage of newly-isolated probiotic strains can be employed to increase the technical production of probiotics for poultry nutrition. The results are expected to be further applied for large-scale manufacturing of these probiotic *Lactobacillus* strains.

## Figures and Tables

**Figure 1 molecules-24-03286-f001:**
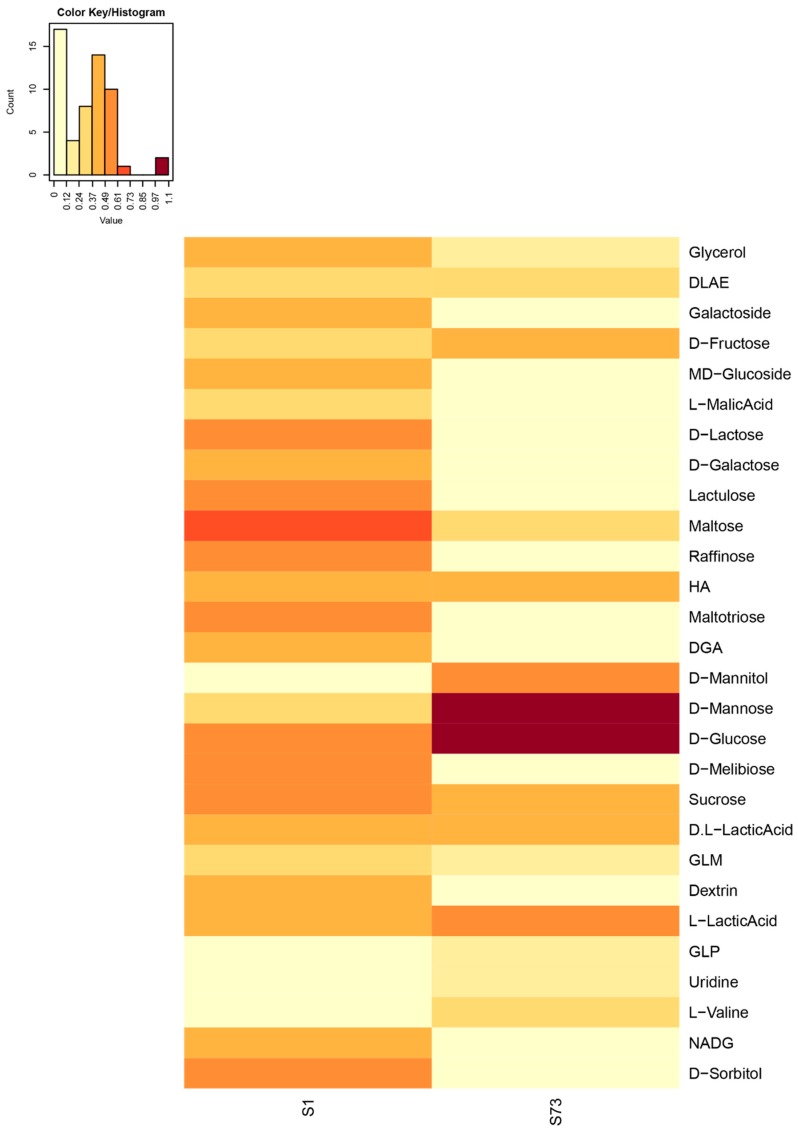
Metabolic fingerprint of the probiotic *Lactobacillus* strains. DLAE = D-Lactic Acid Methyl Ester. HA = α- Hydroxybutyric Acid. DGA = D-Galacturonic Acid. GLM = Glycyl-L-Methionine. GLP = Glycyl-L-Proline. NADG = N-Acetyl-D-Glucosamine. S1 = *L. salivarius*. S73 = *L.agilis*.

**Figure 2 molecules-24-03286-f002:**
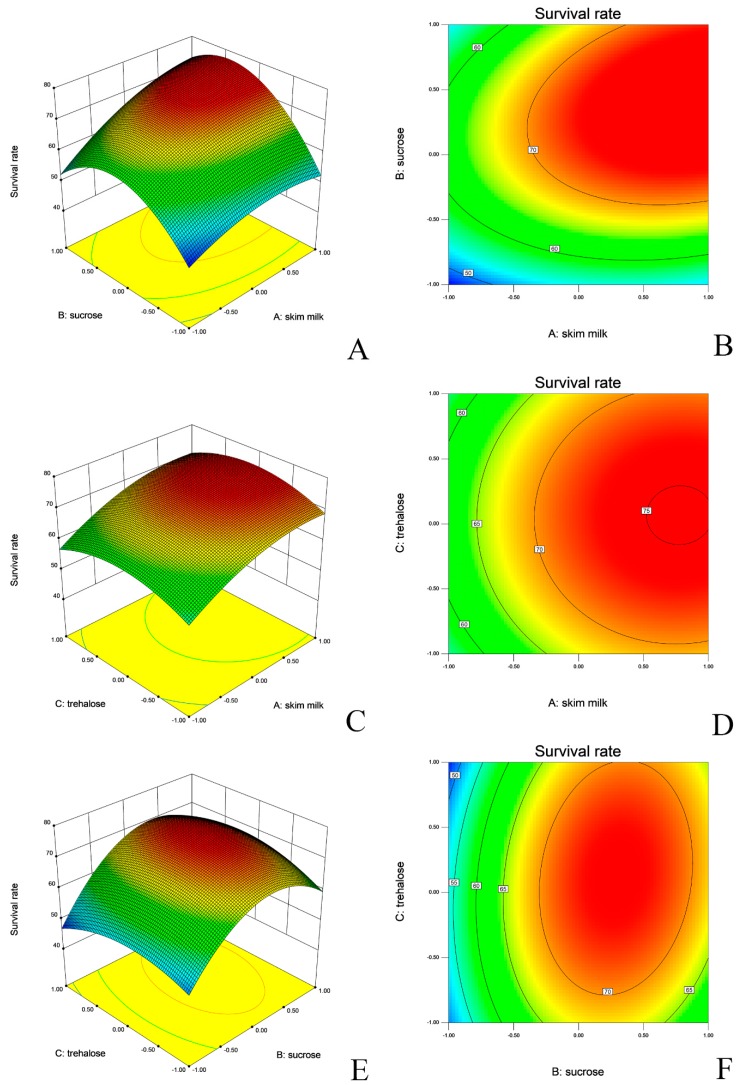
Response surface and contour plots depicting *L. salivarius* viability after lyophilization. (**A**,**B**): skim milk vs sucrose. (**C**,**D**): skim milk vs. trehalose. (**E**,**F**): sucrose vs. trehalose.

**Figure 3 molecules-24-03286-f003:**
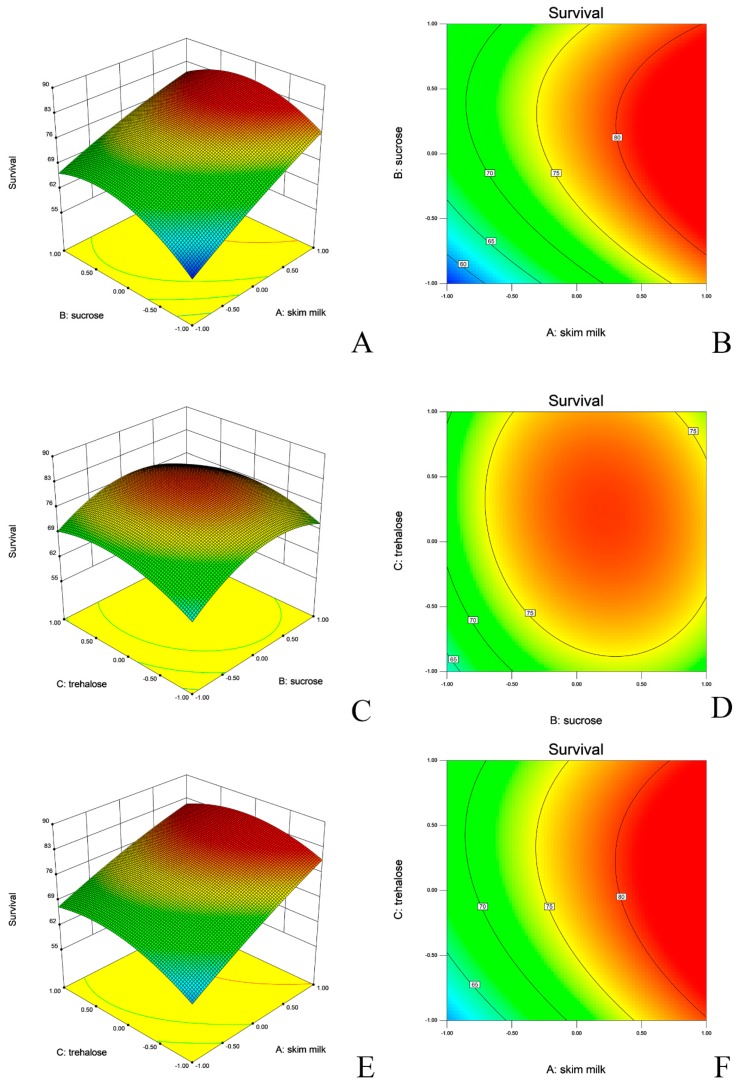
Response surface and contour plots depicting *L. agilis* viability after lyophilization. (**A**,**B**): skim milk vs. sucrose. (**C**,**D**): sucrose vs. trehalose. (**E**,**F**): skim milk vs. trehalose.

**Figure 4 molecules-24-03286-f004:**
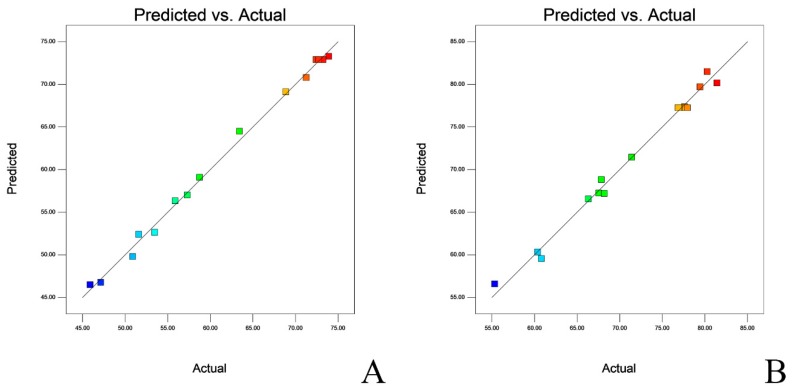
Linear plot fitting predicted vs. actual viability of lactobacilli. (**A**): *L. salivarius*. (**B**): *L. agilis.*

**Table 1 molecules-24-03286-t001:** Biomass of the probiotic strains in media supplemented with different additional substrates at different time points [log CFU/mL].

	12 h		24 h		48 h	
	*L. salivarius*	*L. agilis*	*L. salivarius*	*L. agilis*	*L. salivarius*	*L. agilis*
Sucrose	9.22 ± 0.02 *	9.08 ± 0.02 *	8.94 ± 0.05 *	8.82 ± 0.06 *	8.67 ± 0.12	8.11 ± 0.06 *
Maltose	9.08 ± 0.05 *	9.11 ± 0.07	8.74 ± 0.12 *	8.74 ± 0.12 *	8.26 ± 0.13 *	8.1 ± 0.12 *
Mannitol	9.04 ± 0.11	9.02 ± 0.09	8.75 ± 0.19 *	8.54 ± 0.07 *	8.45 ± 0.09 *	8.21 ± 0.09 *
Mannose	9.1 ± 0.04 *	9.2 ± 0.06 *	8.61 ± 0.03 *	8.92 ± 0.05 *	8.49 ± 0.04 *	8.42 ± 0.1 *
Sorbitol	9.18 ± 0.06 *	9.17 ± 0.03 *	8.88 ± 0.03 *	8.88 ± 0.04	8.48 ± 0.11 *	8.22 ± 0.06 *
Melibiose	9 ± 0.06	9.03 ± 0.03	8.96 ± 0.05	8.52 ± 0.04 *	8.63 ± 0.06	8.27 ± 0.1 *
MRS contol	8.86 ± 0.1	8.98 ± 0.04	9.07 ± 0.07	9.13 ± 0.02	8.65 ± 0.04	8.65 ± 0.07

* = *p* < 0.05. MRS control: control medium (de Man, Rogosa, and Sharpe medium).

**Table 2 molecules-24-03286-t002:** Coefficient estimates and ANOVA (Analysis of variance) analysis of the quadratic model for lactobacilli survival during the lyophilization process.

	Variables	Coefficient Estimates (± Standard Error)	F-Value	*p* Value	Model Significance	R^2^
*L. salivarius*	Intercept	72.9 ± 0.4	233.22	<0.0001	<0.0001 **	0.9924
	Skim milk	6.64 ± 0.32	430.95	<0.0001		
	X_2_	6.76 ± 0.32	446.4	<0.0001		
	X_3_	0.59 ± 0.32	3.41	0.1071		
	Skim milk, sucrose	3.69 ± 0.45	66.4	<0.0001		
	X_1_X_3_	0.24 ± 0.45	0.29	0.6063		
	X_2_X_3_	2.11 ± 0.45	21.72	0.0023		
	X_1_^2^	−4.2 ± 0.44	90.93	<0.0001		
	X_2_^2^	−12.49 ± 0.44	802.54	<0.0001		
	X_3_^2^	−5.37 ± 0.44	148.45	<0.0001		
*L. agilis*	Intercept	77.26 ± 0.52	82.44	<0.0001	<0.0001 **	0.9786
	X_1_	8.6 ± 0.41	440.12	<0.0001		
	X_2_	3.19 ± 0.41	60.54	0.0001		
	X_3_	2.37 ± 0.41	33.46	0.0007		
	X_1_X_2_	−1.79 ± 0.58	9.58	0.0174		
	X_1_X_3_	−1.47 ± 0.58	6.45	0.0387		
	X_2_X_3_	−1.06 ± 0.58	3.36	0.1095		
	X_1_^2^	−1.01 ± 0.56	3.23	0.1155		
	X_2_^2^	−6.07 ± 0.56	115.44	<0.0001		
	X_3_^2^	−4.23 ± 0.56	56.11	0.0001		

X_1_ = skim milk. X_2_ = sucrose. X_3_ = trehalose. ** = *p* < 0.01.

**Table 3 molecules-24-03286-t003:** Optimum process and validation experiment results at a 95% confidence interval.

Response Viability	Target	Predicted Results	Standard Deviation	95% PI Low	95% PI High
*L. salivarius*	Maximized	76.19	3.91	65.54	86.83
*L. agilis*	Maximized	84.77	1.16	81.56	87.97

PI = Prediction interval.

**Table 4 molecules-24-03286-t004:** Survival of the probiotic *L. salivarius* and *L. agilis* during the storage [%].

	*L. salivarius*								*L. agilis*							
	Without Protectants				With Protectants				Without Protectants				With Protectants			
	20 °C		4 °C		20 °C		4 °C		20 °C		4 °C		20 °C		4 °C	
BM	9.01 ± 0.04	100.00%	9.00 ± 0.02	100.00%	9.01 ± 0.02	100.00%	9.00 ± 0.04	100.00%	9.02 ± 0.00	100.00%	9.00 ± 0.03	100.00%	9.01 ± 0.03	100.00%	9.00 ± 0.01	100.00%
DPM0	8.97 ± 0.01	91.56%	8.98 ± 0.01	95.33%	9 ± 0.02	97.74%	8.99 ± 0.03	98.00%	9.00 ± 0.00	97.11%	9.00 ± 0.01	100.67%	9.01 ± 0.00	99.02%	9.01 ± 0.01	101.00%
DPM1	8.97 ± 0.04	90.58%	8.98 ± 0.02	95.00%	8.99 ± 0.05	95.48%	8.99 ± 0.03	98.00%	9.00 ± 0.01	96.46%	9.00 ± 0.03	100.33%	9.01 ± 0.01	99.35%	9.01 ± 0.02	101.00%
DPM2	8.96 ± 0.03	89.29%	8.97 ± 0.04	93.33%	8.99 ± 0	95.16%	8.99 ± 0.01	98.67%	8.99 ± 0.04	94.53%	9.00 ±0.04	100.00%	9.00 ± 0.02	97.07%	9.01 ± 0.01	101.00%
DPM3	8.95 ± 0.03	87.34%	8.96 ± 0.04	91.33%	8.98 ± 0.02	93.55%	8.99 ± 0.02	97.00%	8.99 ± 0.02	93.89%	8.99 ± 0.04	98.67%	9.00 ± 0.00	97.07%	9.00 ± 0.03	99.00%
DPM4	8.96 ± 0.02	88.31%	8.96 ± 0.03	91.33%	8.99 ± 0.01	93.55%	8.99 ± 0.01	97.33%	8.99 ± 0.01	94.86%	8.99 ± 0.00	98.00%	8.99 ± 0.03	96.74%	9.00 ± 0.01	99.34%
DPM15	8.91 ± 0.01a	78.90%	8.92 ± 0.03ab	83.67%	8.96 ± 0.02b	89.03%	8.97 ± 0.01b	94.33%	8.95 ± 0.01a	85.21%	8.96 ± 0.02ab	91.00%	8.99 ± 0.01b	95.44%	8.99 ± 0.01b	98.34%
DPM28	8.64 ± 0.05a	42.86%	8.83 ± 0.03b	67.10%	8.84 ± 0.02b	67.74%	8.93 ± 0.01c	85.33%	8.71 ± 0.04a	49.84%	8.85 ± 0.01a	70.33%	8.91 ± 0.02b	79.48%	8.95 ± 0.02b	88.37%

BM = before mixing. DPM = day-post-mixing. a, b, c = significantly different within a row.

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
