# Peer review of "Optimization of Production Parameters for Probiotic Lactobacillus Strains as Feed Additive"

_molecules, 2019, doi:10.3390/molecules24183286_

Round 1
Reviewer 1 Report
Authors Ren et al. have characterized two broiler-derived probiotic Lactobacillus strains to be used for poultry nutrition. The work reveals to be appealing since it represents an original contribute to feed industry to deepen knowledge as regards the improvement of the technical production of probiotics for poultry nutrition. The present manuscript is of high scientific quality content, clear to the reader, and, overall, well written. English grammar and typing mistakes should be checked throughout the manuscript.
Author Response
Dear Reviewer
We would like to thank you all for the thoughtful comments and constructive suggestions, which help to improve the quality of this manuscript. We have carefully reviewed the comments of reviewers and incorporated the necessary changes in accordance with comments in a way of point-by-point. We hope this current version would meet the publication criteria. Please refer to our response as follows.
Comments and Suggestions for Authors
Authors Ren et al. have characterized two broiler-derived probiotic Lactobacillus strains to be used for poultry nutrition. The work reveals to be appealing since it represents an original contribute to feed industry to deepen knowledge as regards the improvement of the technical production of probiotics for poultry nutrition. The present manuscript is of high scientific quality content, clear to the reader, and, overall, well written. English grammar and typing mistakes should be checked throughout the manuscript.
Answer: Thanks for your valuable comments. The English writing has been improved throughout the manuscript and the mistakes have been corrected.
Reviewer 2 Report
In this study the authors performed the characterization of Lactobacillus salivarius and Lactobacillus agilis strains, isolated from broilers, in relation to substrate utilization, biomass production, oxygen tolerance, cryoprotective agents and feed storage. Based on the results, they concluded that there is a potential for large-scale production of probiotics as a poultry feed additive. The study is scientifically sound and the manuscript is well written. Minor comments are the following:
- Line 64 – 65: The authors stated that Lactobacillus species were isolated in a previous study. Has it been published? Please add the reference;
- Lactobacillus genus and species should be in italics in line 64, figure 1 legend, and Table 4 legend);
- Delete lines 164 – 166 (Discussion section) and 255 – 266 (Materials and Methods section);
- Line 285, please change ODnm590 to OD590nm;
- Please the authors should revise the symbol for degrees Celsius in the Material and Methods section.
Author Response
Dear Reviewer
We would like to thank you for the thoughtful comments and constructive suggestions, which help to improve the quality of this manuscript. We have carefully reviewed the comments and incorporated the necessary changes in accordance with comments in a way of point-by-point. We hope this current version would meet the publication criteria. Please refer to our response as follows.
Comments and Suggestions for Authors
In this study the authors performed the characterization of Lactobacillus salivarius and Lactobacillus agilis strains, isolated from broilers, in relation to substrate utilization, biomass production, oxygen tolerance, cryoprotective agents and feed storage. Based on the results, they concluded that there is a potential for large-scale production of probiotics as a poultry feed additive. The study is scientifically sound and the manuscript is well written. Minor comments are the following:
- Line 64 – 65: The authors stated that Lactobacillus species were isolated in a previous study. Has it been published? Please add the reference;
Answer: The screening and characterization study is documented in another manuscript that is currently under review (Frontiers Microbiol, 1st revision). We added a statement (unpublished data) to clarify their status (line 71) and added the respective reference in the reference section.
- Lactobacillus genus and species should be in italics in line 64, figure 1 legend, and Table 4 legend);
Answer: Many thanks for the comments, we have revised the format accordingly and checked it throughout the manuscript.
- Delete lines 164 – 166 (Discussion section) and 255 – 266 (Materials and Methods section);
Answer: We are sorry for this mistake. The irrelevant texts have been removed accordingly.
- Line 285, please change ODnm590 to OD590nm;
Answer: Thanks for correction. We have already changed ODnm590 into OD590nm in line 296.
- Please the authors should revise the symbol for degrees Celsius in the Material and Methods section.
Answer: Yes, there are some degrees Celsius are in wrong form, we have changed them into ‘°C’. We also add space between numbers and Celsius in tables.
Reviewer 3 Report
I believe that the work is well designed and the subject is of high importance. The methods applied are good and the presentation of the results is fine. However I believe that the authors should introduce more scientific results from the literature in the introduction sector.
Author Response
Dear Reviewer
We would like to thank you for the thoughtful comments and constructive suggestions, which help to improve the quality of this manuscript. We have carefully reviewed the comments and incorporated the necessary changes in accordance with comments in a way of point-by-point. We hope this current version would meet the publication criteria. Please refer to our response as follows.
Comments and Suggestions for Authors
I believe that the work is well designed and the subject is of high importance. The methods applied are good and the presentation of the results is fine. However I believe that the authors should introduce more scientific results from the literature in the introduction sector.
Answer: Thanks for the comments. To make the introduction more comprehensive, we added the statements in line 33-35 (Palma, Zamith-Miranda et al. 2015), 35-36 (Abdou, Hedia et al. 2018) and 55-57 (Huang, Lu et al. 2006) to strengthen the background information about the rationale to choose Lactobacillus as the objective and RSM as the strategy. We also modified some expression in this section.
Abdou, A. M., R. H. Hedia, S. T. Omara, M. A. E.-F. Mahmoud, M. M. Kandil and M. A. Bakry (2018). "Interspecies comparison of probiotics isolated from different animals." Veterinary world 11(2): 227-230.
Huang, L., Z. Lu, Y. Yuan, F. Lü and X. Bie (2006). "Optimization of a protective medium for enhancing the viability of freeze-dried Lactobacillus delbrueckii subsp. bulgaricus based on response surface methodology." Journal of Industrial Microbiology and Biotechnology 33(1): 55-61.
Palma, M. L., D. Zamith-Miranda, F. S. Martins, F. A. Bozza, L. Nimrichter, M. Montero-Lomeli, E. T. A. Marques and B. Douradinha (2015). "Probiotic Saccharomyces cerevisiae strains as biotherapeutic tools: is there room for improvement?" Applied Microbiology and Biotechnology 99(16): 6563-6570.